# Topological Design of a Hinger Bracket Based on Additive Manufacturing

**DOI:** 10.3390/ma16114061

**Published:** 2023-05-30

**Authors:** Baocheng Xie, Xilong Wu, Le Liu, Yuan Zhang

**Affiliations:** Key Laboratory of Advanced Manufacturing Intelligent Technology of Ministry of Education, Harbin University of Science and Technology, Harbin 150080, China

**Keywords:** hinge bracket, topology optimization, lightweight design, additive manufacturing

## Abstract

Topology optimization technology is often used in the design of lightweight structures under the condition that mechanical performance should be guaranteed, but a topology-optimized structure is often complicated and difficult to process using traditional machining technology. In this study, the topology optimization method, with a volume constraint and the minimization of structural flexibility, is applied to the lightweight design of a hinge bracket for civil aircraft. A mechanical performance analysis is conducted using numerical simulations to obtain the stress and deformation of the hinge bracket before and after topology optimization. The numerical simulation results show that the topology-optimized hinge bracket has good mechanical properties, and its weight was reduced by 28% compared with the original design of the model. In addition, the hinge bracket samples before and after topology optimization are prepared with additive manufacturing technology and mechanical performance tests are conducted using a universal mechanical testing machine. The test results show that the topology-optimized hinge bracket can satisfy the mechanical performance requirements of a hinge bracket at a weight loss ratio of 28%.

## 1. Introduction

With the rapid development of science and technology, the civil aviation industry has experienced great progress, and airplanes have become an indispensable means of transportation in people’s daily travel. Research shows that improvements to the flight speed, thrust-to-weight ratio, and aircraft lift can range from 3% to 5% for every 1% reduction in the aircraft weight [1,2,3,4]. The hinge bracket, as the most important component of aircraft doors, plays a vital role in the safety of passengers during the emergency evacuation of an aircraft. Considering the demanding requirement of a high stiffness-to-weight ratio in the aerospace industry, a hinge bracket should have a lightweight design under the condition that the mechanical properties, such as intensity and stiffness, meet the safety requirements.

Currently, topology optimization technology is the main way to achieve a lightweight design. Topology optimization technology is the realization of the on-demand distribution of materials within a component’s structure based on an optimal force transfer path to achieve the most efficient structural performance while providing good economic benefits [5,6]. With the emergence of the homogenization method, continuum topology optimization methods have rapidly been developed, gradually forming into variable density methods, including solid isotropic material with penalization (SIMP) [7,8] and rational approximation of material properties (RAMP) [9,10,11], evolutionary structural optimization (ESO) [12,13], and level set (LS) [14,15]. However, topology-optimized structures are generally complex and difficult to machine using traditional processing technology. Therefore, topology-optimized structures should be designed again to ensure the manufacturability of the structure [16,17,18]. The application of additive manufacturing technology has made it possible to manufacture complex topologically optimized structures [19], which promotes the application of topology optimization technology in the design of lightweight structures.

In this paper, a hinge bracket for civil aircraft was designed to be lightweight using the topology optimization method. To evaluate the mechanical performance of the hinge bracket before and after topology optimization, a stress and deformation analysis of the hinge bracket was conducted using numerical simulation. In addition, the hinge bracket samples before and after topology optimization were prepared using additive manufacturing to verify the usability and rationality of the topology-optimized hinge bracket.

The organization of this thesis is as follows: Section 2 reviews recent research on the combination of topology optimization and additive manufacturing. Section 3 utilizes the variable density method topology optimization technique for the lightweight design of the hinge bracket. Section 4 provides a mechanical analysis of the hinge bracket before and after optimization using simulations and experiments. Section 5 summarizes the research and discusses future research directions.

## 2. Literature Review

Topology optimization can be a good way to design lightweight parts while maintaining their performance, but the complexity of topology-optimized parts makes them significantly more difficult to manufacture. However, with the advent of additive manufacturing, this problem has been solved. Yang et al. [20] proposed an orthotropic anisotropic material and penalized solid orthotropic material with a penalization (SOMP) topology optimization method combined with the 3D printing of manufacturable continuous carbon fiber-reinforced composites (CCFRCs) to determine the structure of the specimens, and verified that the stiffness and stiffness-to-weight ratio of the SOMP specimens improved by 30.0% and 26.3%, respectively. Yap et al. [21] explored the application of selective laser sintering (SLS) printing technology and topology optimization technology in ultralight UAVs and verified the effectiveness of the topology-optimized structures through finite element simulation and 3D printed objects. Riad Ramadani et al. [22] designed and optimized the honeycomb gear body structure using the topology optimization software ProTOp, performed SLM machining using Ti-6Al-4V alloy and verified experimentally that the studied gear body cell-dotted structure can well reduce the mass and vibration of the gear. Naruki Ichihara et al. [23] proposed a method to improve the structural toughness of 3D-printed carbon fiber-reinforced composites with localized dotting using the percentage of the intermediate materials obtained in the topology optimization. Wang et al. [24] proposed a method to generate self-supporting lattice structures in a topology-optimized framework that can effectively generate self-supporting lattice structures with higher mechanical strength. Liu et al. [25] designed a lightweight sandwich aircraft spoiler with a high strength-to-weight ratio and filled the topologically optimized space inside the aircraft spoiler with a 3D kagome lattice by combining topology optimization and lattice structure techniques. The optimized structure was verified to have good mechanical properties via numerical simulation, while the designed model was fabricated using stereolithography appearance (SLA) technology. Tang et al. [26] proposed an optimization method for topology optimization and cross-fill angles in material extrusion-based additive manufacturing, which was verified using microstructural and numerical simulations of 3D printed parts, and it was an effective method for structural optimization. Zeng et al. [27] designed an electric vertical takeoff and landing aircraft with a lightweight design of the structural components based on the variable density method, while some major components were designed using the fused deposition modeling (FDM) method to reduce the weight of the aircraft. Devin P. Anderson et al. [28] used topology optimization techniques to find the optimal distribution of materials for lightweight electrolyzer end plates and fabricated new electrolyzer end plates via electron beam powder bed fusion (EB-PBF). S. Mantovani et al. [29] used topology optimization techniques to design an upright geometry column for the automotive field, with weight minimization as the optimization objective and also fabricated a new structure using laser powder bed fusion (L-PBF).

Specifically, existing studies on the combination of topology optimization and 3D printing can be summarized as shown in Table 1.

Current topology optimization software is based on the variable density method and the new method. The usual ordinary continuum model for topology optimization can be fast and efficient. The topology optimization variable density method can effectively solve the problem of the slow convergence of the numerical solution algorithm and, at the same time, improve the quality of the solution, as well as have other advantages;Topology optimization did not exist before the emergence of 3D printing. Only 3D printing can realize the manufacturing of complex parts with topology optimization, which has caused topology optimization to gain increasing attention, promoting research on the combination of topology optimization and 3D printing.

## 3. Topology Optimization of the Hinge Bracket

### 3.1. Topology Optimization Method

Topology optimization is a method used to optimize the stress transfer path of a structure by reconfiguring materials within the design domain under external loads and constraints. The aim is to achieve a partially optimized system that meets strength, stiffness, and displacement requirements while fulfilling all other criteria [30]. The variable density method is the most typical approach in topology optimization. The variable density method mainly uses the density of the material unit as a design variable. The magnitude of the load applied to each material cell is analyzed considering the boundary conditions, and the density of the content cell is used to reflect the load it is subjected to. By applying an interpolation function to penalize the material unit located in the medium density, in order to further promote the density of the material unit gradually tends to 0 or 1. Finally, the part of the density that tends to 0 is deleted, and the part of the density that tends to 1 is retained so as to achieve the optimal design of the material unit [31,32,33].

In this paper, the variable density method is used as the theoretical basis for the solid isotropic material penalty model. We set the material density as the design variable and the minimization of structural flexibility as the optimization objective while obeying the constraints on the structural volume. Based on these factors, we develop a mathematical model for the topology optimization of a hinged bracket. [34]:

The mathematical model of the SIMP method of the interpolation function is [35]
(1)$$E\left(x_{i}\right)=E_{\min}+{\left(x_{i}\right)}^{p}\left(E_{0}-E_{\min}\right),x_{i}\in \left[0,1\right]$$
where $E\left(x_{i}\right)$ is the elastic modulus after interpolation; $E_{0}$ is the elastic modulus of the solid part of the material; $E_{\min}$ is the elastic modulus of the hole part of the material; $x_{i}$ is the relative density of the unit: when the value is 1, it means there is material, and when it is 0, it means there is no material, i.e., hole; p is the penalty factor.

The design variables can be expressed as
(2)$$\begin{array}{c} find\;X={\left(x_{{}_{1,1}},x_{{}_{1,2}},x_{{}_{1,3}},L,x_{{}_{i,j}}\right)}^{T}\hat{I}R\\ i=1,2,L,n \end{array}$$

The optimization objective can be expressed as
(3)$$\min\;C\left(X\right)=FU=U^{T}KU=\sum_{i=1}^{m}\sum_{j=1}^{n}u^{T}{}_{i,j}k_{i,j}u_{i,j}=\sum_{i=1}^{m}\sum_{j=1}^{n}{\left(x_{i,j}\right)}^{p}u^{T}{}_{i,j}k_{0}u_{i,j}$$

The constraint can be expressed as
(4)$$Subject\;to\;\{\begin{array}{c} KU=F\\ V=fV_{0}=\sum_{i=1}^{m}\sum_{j=1}^{n}x_{i,j}\cdot v_{i,j}\\ 0<x_{\min}\leq x_{i,j}\leq x_{\max}\leq 1 \end{array}$$
where *X* is the cell-relative density vector; C is the flexibility of the structure; *F* is the vector of the load; U is the displacement vector; k is the structural stiffness matrix; $u_{i,j}$ is the cell displacement vector; *K_i,j_* is the cell stiffness matrix; $k_{0}$ is the initial segment element stiffness matrix; $v_{i,j}$ is the cell volume; *V* is the optimized volume; *f* is the retained volume fraction; *V*_0_ is the initial volume; *x*_min_ is the lower limit of the values of the design variables, and *x*_max_ is the upper limit of the values of the design variables; *n* is the number of cells in the subdomain.

Topology-optimized solution process for the variable density method: Firstly, establish a finite element model and perform meshing of the model. Then determine the design area, loads, and constraints of the structure, and set its boundary conditions. Next, determine the design variables, update the dimensions of the smallest members, end the process, and obtain the topology-optimized structure if convergence is achieved; otherwise, continue designing.

### 3.2. Topology Optimization Process of the Hinge Bracket

#### 3.2.1. Hinge Bracket Modeling

The geometricality models the hinge bracket in Creo with a 1:1 ratio, and all initial parameters used in the study are derived from publicly available data on the website. Simplification of the model should be performed before meshing to save computational time. The rounded corners of the hinge bracket are small and prone to distortion when the mesh is divided. Therefore, the rounded corners are simplified to a right-angle structure. The three-dimensional model of the hinge bracket is shown in Figure 1.

#### 3.2.2. Hinge Bracket Load Analysis

The hinge bracket is connected to the primary hinge on one side and to the emergency hatch on the other side of the civil aircraft emergency hatch. Through the analysis of the movement process of the emergency door, it can be known that the hinge bracket and the door body are fixed and connected by bolts, and the bottom bolt hole of the hinge bracket is subject to fixed constraints. In addition, the hinge bracket is also subjected to the force of the main hinges, power assist guides, and other parts. The load applied to the hinge bracket is shown in Figure 2, and the magnitude of the values is shown in Table 2.

### 3.3. Simulation of Hinge Bracket Topology Optimization

In the optimization panel of the analysis module, the previously set optimization area is selected in topology. The topology optimization of the hinge bracket is performed with the volume removal rate of the hinge bracket as the constraint and the minimum flexibility of the hinge bracket as the optimization objective, and the upper limit of material retention is adjusted to 0.3, which means that 70% of the material is removed. However, only considering the constraint of material removal rate may cause the concentrated material in the heavily loaded parts of the topology-optimized hinge bracket. Therefore, it is necessary to introduce member size constraints to control the maximum and minimum member sizes, ensuring that the topology-optimized hinge bracket has multiple paths for force transmission. In general, the member size is usually set between three and six times the meshing cell size. In this paper, the mesh division cell size of the hinge bracket is 2 mm, so the minimum member size is set to 6 mm, and the maximum member size is set to 12 mm.

According to the topology optimization-related parameters mentioned above, the volume fraction iteration curve of the hinge bracket was obtained after completing the topology optimization. When the number of iterations reached 20 times, the volume fraction was relatively stable.

### 3.4. Optimization Results

According to the topology optimization, the density cloud diagram of the hinge bracket is obtained, as shown in Figure 3a. The red color in the figure indicates the area is subject to a higher load, the blue color indicates that it is subject to a lower load, and the blue area part can be removed. The excessive from the blue area to the red area means that the local material is more and more critical to the structural performance.

According to the topologically optimized density cloud diagram of the hinge bracket, it can be seen that the material of the hinge bracket is mainly distributed at the connection locations of the upper face holes of the side plates and the bolt holes of the bottom flat plate. Therefore, the most effective way to ensure the stiffness requirements of the hinge bracket is to arrange reinforcement plates at each connection. In addition, since the hinge bracket needs to be connected to other parts, it is necessary to ensure the accuracy of the position of each hole in the hinge bracket, so the main improvement was made to the connection of the holes in the upper part of the plate. On the basis of ensuring the bottom and side holes of the hinge bracket remain unchanged, the part of the density cloud less than 0.3 is removed, and the part of the density cloud greater than 0.3 is retained to obtain the optimized hinge bracket, as shown in Figure 3b.

Further processing was carried out on the topology-optimized hinge bracket. The new hinge bracket, as shown in Figure 4, achieved a 28% reduction in material and significantly reduced the weight of the component compared to traditional hinge brackets.

## 4. Results and Discussion

### 4.1. Post-Optimization Simulation Analysis

#### 4.1.1. Finite Element Simulation

Before the tensile experiment, the results are predicted by finite element analysis. In order to verify the feasibility of the topology-optimized design results, the topology-optimized hinge structure also needs to be calibrated to analyze whether it meets the design and usage requirements.

The mesh of the hinge bracket was divided by HyperMesh, and the numerical simulation of the hinge bracket before and after optimization was performed by using the finite element analysis software ANSYS Workbench 2019. A tetrahedral mesh with an average size of 2 mm was used for meshing the hinge bracket. As shown in Figure 5, the number of nodes was 81,010, and the number of cells was 324,428, which is considered appropriate. Then, the mesh quality was checked, and the percentages of check items Jacobi, warpage angle, aspect ratio, skew angle, maximum internal angle of the tetrahedron, and minimum internal angle of the tetrahedron were less than or equal to 1%, so the mesh division of this hinge bracket is reasonable. The boundary conditions were set according to the numerical size and the orientation of the hinge bracket diagram in Table 2 to obtain information about the magnitude of deformation, stress, and strain of the hinge bracket before and after optimization, as shown in Figure 5. The material set here was 7050 aluminum alloy, a high-strength, stress, and corrosion-resistant high-performance aerospace aluminum alloy widely used in aircraft structural parts. Its properties are shown in Table 3. This simulation did not take into account any damage caused to the mechanism since no fracture or damage was expected to occur in accordance with the boundary condition settings.

#### 4.1.2. Simulation Results

According to the deformation diagram of the hinge bracket in Figure 6, it can be seen that the maximum deformation occurs at the stress point of hole B. The maximum stress appears below the stress point of hole B. The maximum strain occurs at the connection between hole B and the beam structure.

Before and after optimization, the maximum deformation of the hinge bracket was 0.017 mm and 0.024 mm, respectively. Structural optimization led to a reduction in the material used, which caused a decrease in the structure’s stiffness and increased the deformation of the optimized hinge bracket. Although parts with a density of less than 0.3 were deleted from the topology optimization’s density cloud diagram, these parts still offered partial support for the structure. The new hinge bracket obtained by topology optimization exhibited partial stress concentration that increased the stress on the optimized hinge bracket. The pre-optimized hinge bracket’s maximum stress under load was 57.84 MPa, which is much lower than the allowable stress of 294 MPa, satisfying its strength requirements. However, the optimized hinge bracket’s maximum stress was 69.4 MPa, which has increased compared to before optimization but still meets the strength requirements. Due to the optimized structure, the load on the structure distributes to larger deformation, causing an increase in the strain of the optimized hinge bracket for the same load. The total strains before and after optimization were 0.00029 and 0.0004, respectively, and simulation results demonstrate that the topologically optimized hinge bracket still meets its operating conditions. Furthermore, the weight of the optimized hinge bracket has been reduced by 28%.

### 4.2. Experimental Analysis after Optimization

#### 4.2.1. Fabrication of the Model

The experimental model was printed and processed by the Lite-600HD light-curing 3D printer of Shanghai Liantai Company, and the equipment processing parameters are shown in Table 4. This printer uses the principle of laser scanning curing resin to print the model, and the generated hinge bracket is well-formed without the advantages of hollow depression and deformation. Considering the cost of 3D printing and the existing experimental conditions, SLA printing was used. The material used in the experiment is white resin UTR8220 with a tensile strength of 51.21 Mpa and a density of 1.13 g/cm^3^, which is a super tough, high strength, and high hardness ABS material. The cured material has a smooth surface and is usually used in the production of molds, electrical parts, etc.

The printed hinge bracket was removed from the bottom, and the support inside the round hole was with a spatula, followed by an alcohol cleaning solution to clean off the resin adhering to the surface of the part, and finally cleaned and dried, then put it into the UV oven for the second light curing operation. After the above operation, the 3D-printed hinge bracket was obtained, as shown in Figure 7.

#### 4.2.2. Tensile Experiment Test

The tensile test was conducted on an INSTRON 3328 universal mechanical testing machine. However, due to the inadequacy of the experimental conditions, only the two most stressed bolt holes underwent force experiments using the universal machine. The universal testing machine and the test procedure are shown in Figure 8, and the tensile deformation rate is 8 × 10^−3^ mm/s.

#### 4.2.3. Experimental Results

Figure 9 shows the load-displacement curves of the hinge bracket before and after optimization. From Figure 9, the original hinge bracket’s degree of load and displacement change during its initial tensile stage was slower than the corresponding quantities for the optimized hinge bracket. The damage load of the original hinge bracket is 1.5 KN, the damage load of the optimized hinge bracket is 1.2 KN, and the difference between the damage loads of the two is not very large. Due to the limited experimental conditions, only two stress points were used for the tensile experiments, which did not correspond to the simulated experimental values, but the overall trend of the values was approximately the same.

## 5. Conclusions

In this paper, a topology optimization model with the minimum structural flexibility as the objective function, the density of the material as the design variable, and SIMP as the interpolation function of the variable density method is established, and a new hinge bracket model is obtained by redesigning the hinge bracket through the analysis of topology optimization of the hinge bracket. The hinge bracket before and after optimization was compared and analyzed from three perspectives of deformation, stress, and strain using the finite element method. The results showed that the weight of the designed hinge bracket was reduced by 28%. In addition, the maximum stress of the new hinge bracket is 69.4 MPa, which is still less than its yield strength and meets its usage requirements. Finally, we made models of the two hinge brackets using additive manufacturing technology, tested them, and concluded that the difference in failure load between them was not very large. However, we acknowledge that the hinge bracket designed in this case is not optimal and needs further exploration in the future. It is necessary to fabricate models using 7050 aluminum alloy material for testing. However, there is still some way to go to achieve these, and the authors may continue their research on the optimization of the hinge bracket in the future.

## Figures and Tables

**Figure 1 materials-16-04061-f001:**
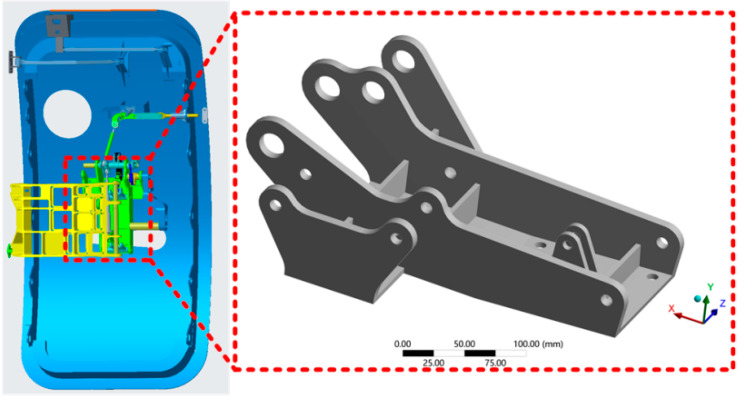
Three-dimensional model of the hinge bracket.

**Figure 2 materials-16-04061-f002:**
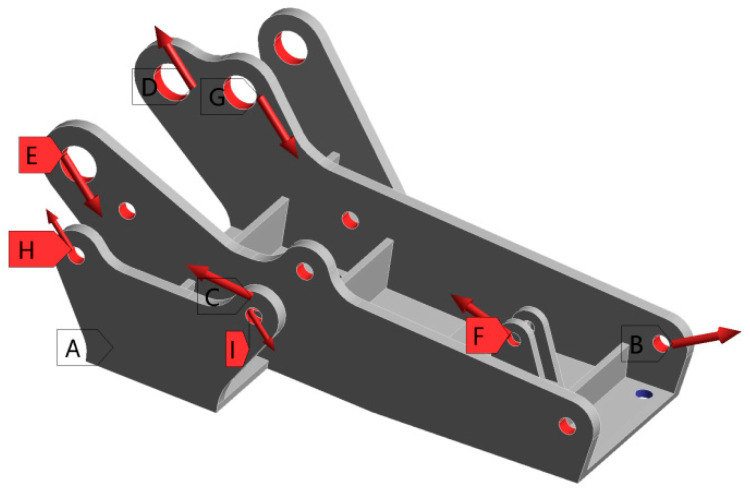
Load application diagram of hinge bracket.

**Figure 3 materials-16-04061-f003:**
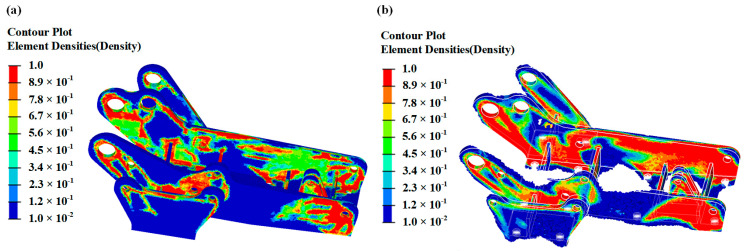
(**a**) Density cloud of hinge bracket topology optimization; (**b**) hinge bracket optimization.

**Figure 4 materials-16-04061-f004:**
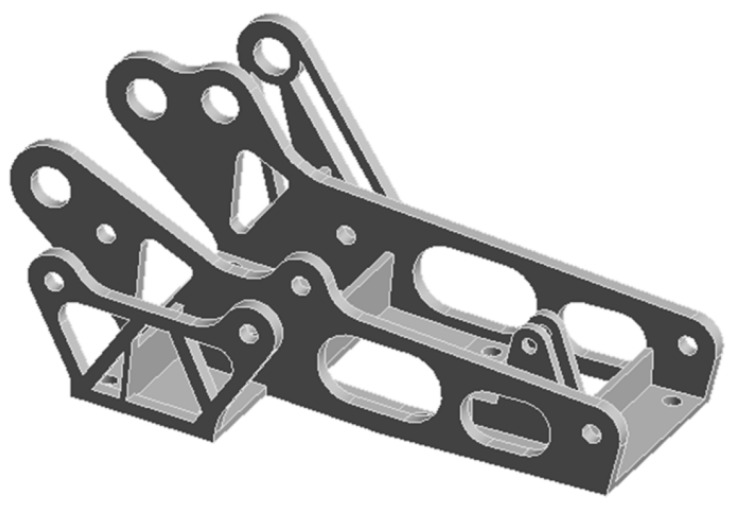
Topology optimization model of the hinge bracket.

**Figure 5 materials-16-04061-f005:**
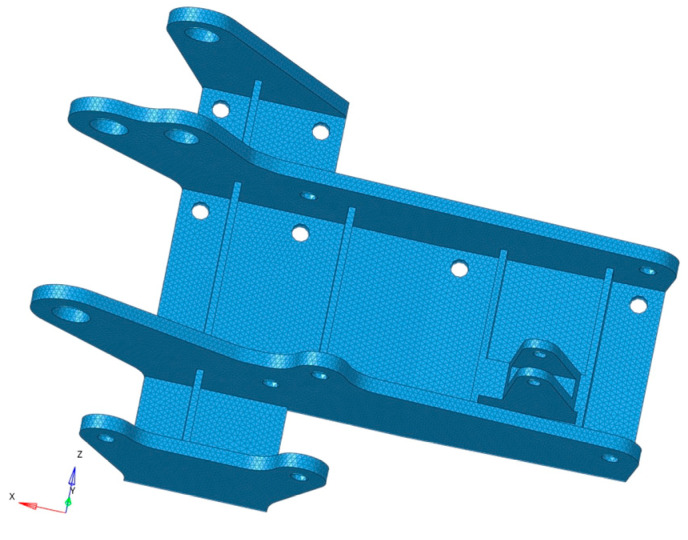
Hinge bracket meshing.

**Figure 6 materials-16-04061-f006:**
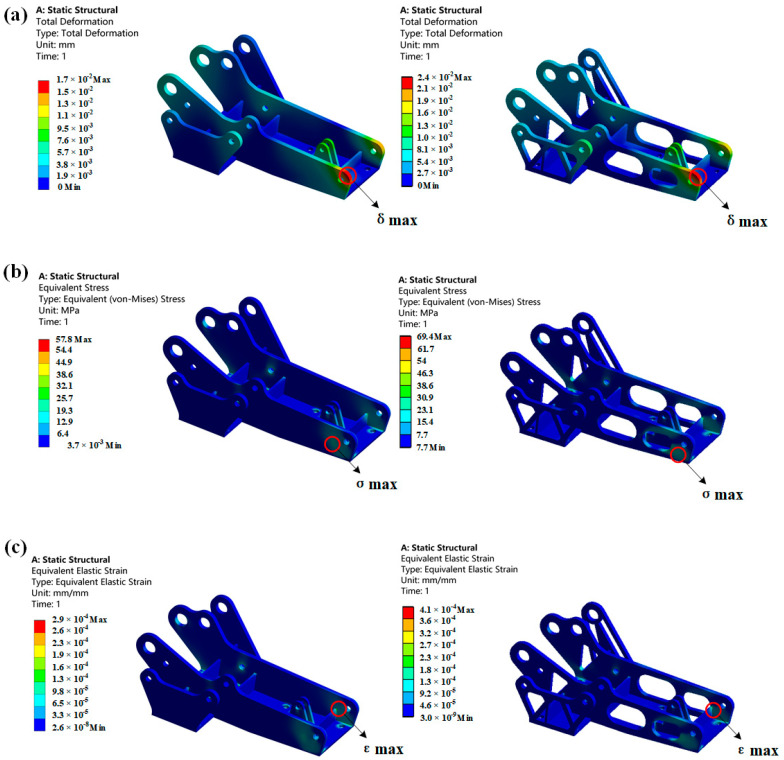
Simulation diagram of hinge bracket before and after optimization: (**a**) deformation diagram; (**b**) stress diagram; (**c**) strain diagram.

**Figure 7 materials-16-04061-f007:**
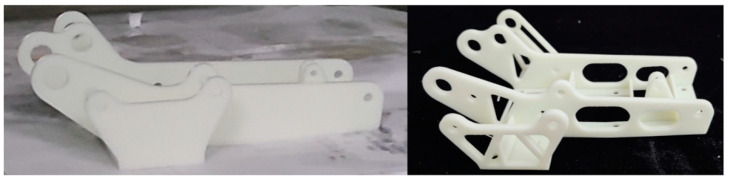
Sample diagram of the optimized front and rear hinge bracket.

**Figure 8 materials-16-04061-f008:**
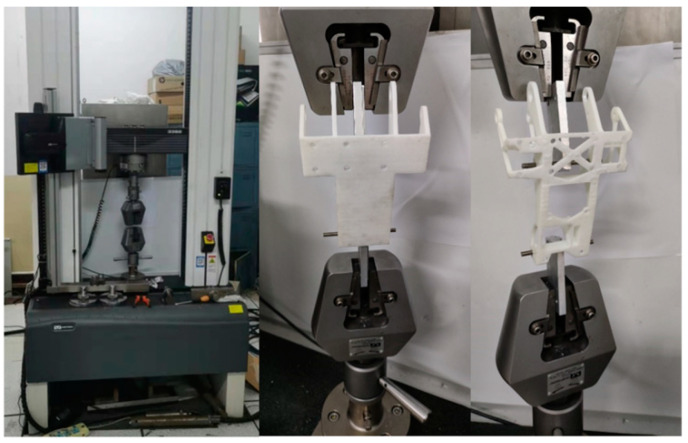
Universal mechanical testing machine and test process.

**Figure 9 materials-16-04061-f009:**
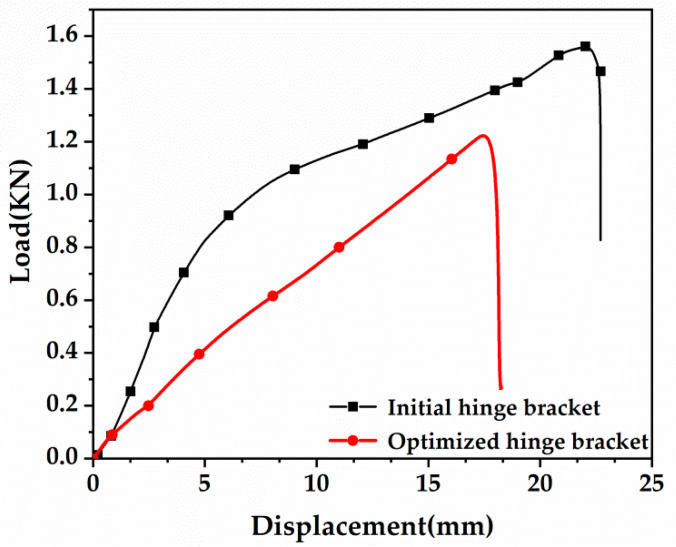
Load-displacement curve of hinge bracket before and after optimization.

**Table 1 materials-16-04061-t001:** Literature review on topology optimization and selection of AM printers.

Author(s)	Optimization Objectives and Structure	Topology Optimization Methods	AM Types
Yang et al. (2022) [20]	Overall Flexibility	SOMP	FDM
Yap et al. (2023) [21]	The overall flexibility of the drone	Variable density method	SLS
Riad Ramadani et al. (2021) [22]	Gear quality	Variable density method	SLS
Naruki Ichihara et al. (2023) [23]	Toughness of the structure	SOMP	FDM
Wang et al. (2023) [24]	Strength of lattice structure	Variable density method	SLA
Liu et al. (2019) [25]	Spoiler maximum stiffness	Variable density method	SLA
Tang et al. (2022) [26]	Stiffness of the structure	Homogenization method	FDM
Zeng et al. (2022) [27]	Weight of electric vertical takeoff and landing aircraft	SIMP	FDM
Devin P. Anderson et al. (2022) [28]	Electrolytic tank end plate weight	Variable density method	EB-PBF
S Mantovani et al. (2021) [29]	The weight of the car column	Variable density method	L-PBF

**Table 2 materials-16-04061-t002:** Loads of hinge brackets.

Load Position	Fx (N)	Fy (N)	Fz (N)	Distribution Area (mm^2^)
A	Fix	Fix	Fix	251
B	−2279	1140	704	251
C	2279	695	−704	251
D	1200	1050	0	251
E	−496	−498	0	503
F	746	232	0	503
G	−220	−246	0	503
H	200	200	0	251
I	−200	−200	0	251

**Table 3 materials-16-04061-t003:** 7050 aluminum alloy properties.

Materials	7050 Aluminum Alloy
Density (kg/m^3^)	2823
Poisson’s ratio	0.346
Young’s modulus (N/m^2^)	7 × 10^10^
Tensile strength (Mpa)	510
Yield strength (Mpa)	441

**Table 4 materials-16-04061-t004:** Lite-600HD light-curing printer device processing parameters.

Equipment Model	Lite-600HD
Forming range	600 × 600 × 400 mm
Forming accuracy	L < 100 mm; ±0.1 mm L > 100 mm; ±0.1% × L
Print layer thickness	0.05–0.25 mm
*Z*-axis positioning accuracy	≤±8 μm
Scanning speed	18 m/s(Max) 8–15 m/s (Typical)
Light spot diameter	0.12–0.80 mm

## Data Availability

Not applicable.

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
