# Peer review of "Topological Design of a Hinger Bracket Based on Additive Manufacturing"

_materials, 2023, doi:10.3390/ma16114061_

Round 1

Author Response

Dear Reviewers:

Thank you for your letter and the reviewers' comments on our manuscript. Those comments are all valuable and very helpful in revising and improving our paper, as well as the important guiding significance to our researches. We have studied comments carefully and have revised the content of the manuscript which we hope meet with approval. Revised portion is marked in red in the paper. The main revisions in the paper and to the reviewers' comments are in the attachment.

Reviewer 2 Report

The manuscript is well organized and written. The paper is of appropriate length. The title and abstract are satisfactory. Finally, the Conclusions part does a good job in wrapping up the paper by summarizing the main findings. However, here are some comments and suggestions which can help improve the quality of the manuscript.

·       An image of the meshed components can be included in the manuscript

·       Were any mesh refinement studies conducted?

·       The quality of the images needs improvement

·       Under heading 3.1.2. Simulation results, “According to the deformation diagram of the hinge bracket in Figure 6,”. The figure number should be Figure 5.

·       Why was the deformation higher before the optimization? The authors are required to provide suitable reasons. Similarly, reasons for difference in the observed stress and strain should be explained.

·       The authors conducted the FEM simulations of the bracket made of aluminium alloy 7075. However, for the experiments, bracket made of UTR8220 resin was used. Since the material properties are entirely different, can such an optimization and comparison be carried out?

Some minor grammatical corrections needed.

Author Response

(The authors gave the same response as above.)

Reviewer 3 Report

In this work, Authors used topology optimization approach to change the design of a hinger bracket used in the aerospace applications. This is an interesting topic in general for the people in this field, currently there are many parts manufactured using AM processes, however, in many cases the geometry of the parts could be improved. In other words, AM processes provide much more freedom regarding the conventional methods, to manufactures lighter and more efficient parts. I have the following comments on the submitted manuscript: 

- Authors used very simple methods for optimizing the design. Currently there are different software in the market which already provide this calculations with further details. What are the advantage of your calculations regarding to commercial software like ntopology,.

- Table 1 lists some loads on the bracket. How Authors define these forces? these are coming from service conditions?

- Authors conducted the simulations on Al alloy, and finally print the part in non-metal material. What is the point of this printing? I think this can only act as prototype to help you to have an idea about your design. but Authors in the next step are taking mechanical tests from these parts. one point is the difference between simulations and experimental results, and the other point is the materials performance and we know that metallic materials deformation mechanisms are completely different. This is why we see this different plastic deformation in the one design. In addition, the difference in the mechanical test might also come from defects in the parts and it needs to be confirmed by several test. But the best is to find a LPBF machine and try it in Aluminium.

- the manuscript is submitted to the Materials journal, but I think there is almost no discussion regarding the materials and the manuscript is 100% on design.

Author Response

(The authors gave the same response as above.)

Round 2

Reviewer 3 Report

Dear Authors,

Many thanks for your answers. Honestly most of the answers does not change too much the manuscript condition. The manuscript is submitted to the state of art of AM issue, and what is presented here is using a module in a software to optimize the design of a part for additive manufacturing where finally part also is printed in another material.you can see example of some similar works in publications or commercial software:

https://www.mmsonline.com/cdn/cms/uploadedFiles/Topology-Optimization-of-an-Additive-Layer-Manufactured-Aerospace-Part.pdf

https://www.sciencedirect.com/science/article/pii/B9780128140628000054

https://www.sciencedirect.com/science/article/pii/S0010448516300951

https://www.sciencedirect.com/science/article/pii/S1000936119303358

https://www.sciencedirect.com/science/article/pii/S0278612514000831

Listing these studies does not mean we do not need further work on this topic, as I believe this is an important point for AM process but probably some small novelties would make it more interesting for readers, these could be for example increasing printability of the part or finally print and test real parts.

The addition of density of a material in the text does not mean that you covered the materials story, my point was to discuss and show the effect of the material microstructure, the effect of the process parameters on the performance of your part.

Another important point in topology optimization is to consider the effect of the support structure, Many parts are coming with advanced designs but it would be difficult ti print them, with the process and material you printed probably it is not an issue but the challenges are coming when you go to a metal printer.

I put acceptance for the current manuscript, Finally editor can decide  the final decision. I understand in the current status, the required modifications might be too much for Authors to adapt for improvement of this manuscript.